# Any-to-Bokeh: Arbitrary-Subject Video Refocusing with Video Diffusion Model

**Yang Yang**[1,2*]     **Siming Zheng**[2*]     **Qirui Yang**[2]     **Jinwei Chen**[2]     **Boxi Wu**[1†]
**Xiaofei He**[1]     **Deng Cai**[1]     **Bo Li**[2]     **Peng-Tao Jiang**[2†]
[1]Zhejiang University     [2]vivo BlueImage Lab, vivo Mobile Communication Co., Ltd.

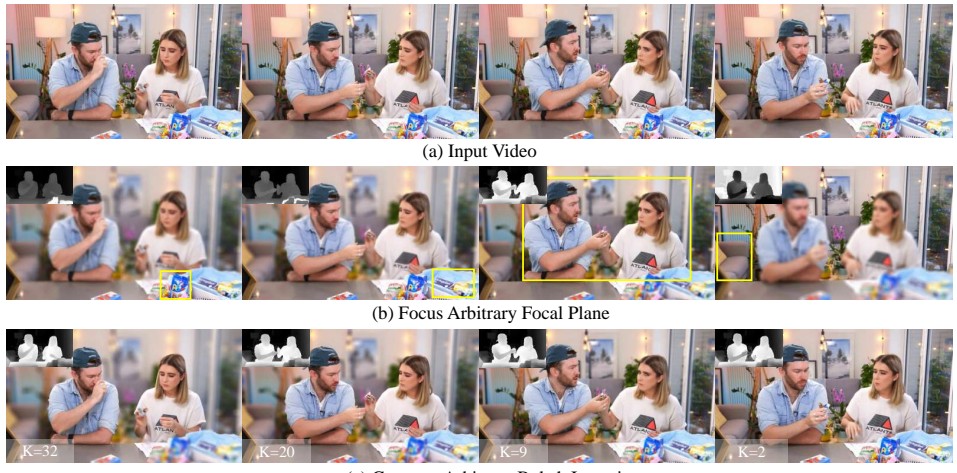

(a) Input Video

(b) Focus Arbitrary Focal Plane

(c) Generate Arbitrary Bokeh Intensity

Figure 1: **Any-to-Bokeh enables users to customize the focal plane and adjust bokeh intensity**. The yellow box indicates the focal plane, and the grayscale values in the image represent the distance to the focal plane, with higher values indicating closer proximity. $K$ represents the bokeh intensity.

## Abstract

Diffusion models have recently emerged as powerful tools for camera simulation, enabling both geometric transformations and realistic optical effects. Among these, image-based bokeh rendering has shown promising results, but diffusion for video bokeh remains unexplored. Existing image-based methods are plagued by temporal flickering and inconsistent blur transitions, while current video editing methods lack explicit control over the focus plane and bokeh intensity. These issues limit their applicability for controllable video bokeh. In this work, we propose a one-step diffusion framework for generating temporally coherent, depth-aware video bokeh rendering. The framework employs a multi-plane image (MPI) representation adapted to the focal plane to condition the video diffusion model, thereby enabling it to exploit strong 3D priors from pretrained backbones. To further enhance temporal stability, depth robustness, and detail preservation, we introduce a progressive training strategy. Experiments on synthetic and real-world benchmarks demonstrate superior temporal coherence, spatial accuracy, and controllability, outperforming prior baselines. This work represents the first dedicated diffusion framework for video bokeh generation, establishing a new baseline for temporally coherent and controllable depth-of-field effects. Project page is available at this website.

## 1 Introduction

Recent advances in diffusion models have significantly advanced camera simulation, enabling controllable geometric transformations such as lens movement, zooming, and panning (Teng et al., 2023; Shi et al., 2024; Wu et al., 2024; Yin et al., 2023). These developments demonstrate their strong potential for reproducing realistic optical effects beyond traditional rendering pipelines.

---

† Corresponding author. * Equal contribution.

Building on these advances, diffusion-based methods have recently been extended to image bokeh rendering (Fortes et al., 2025; Zhu et al., 2025). By leveraging generative priors, these methods achieve visually convincing blur transitions and highlight the feasibility of applying diffusion models to depth-aware, image-level optical effects. However, these methods focus on image inputs, with limited temporal modeling, so their applicability to videos is constrained.

Compared to image-based bokeh, video bokeh poses additional challenges remains limited in terms of temporal coherence and controllability. The limitations are two-fold. (i) Frame-by-frame extensions of image methods (Sheng et al., 2024; Peng et al., 2022b;a; Lijun et al., 2018; Zhu et al., 2025) lack explicit temporal modeling and rely on imperfect depth estimates. Together with stochastic noise accumulation in multi-step diffusion, this often causes **temporal flickering** and unstable blur boundaries, degrading video fidelity. (ii) Although recent video generative models (Blattmann et al., 2023) provide strong 3D priors and cross-frame consistency, their bokeh-like effects are largely implicit and lack explicit control over the **focal plane** and **bokeh intensity**, limiting realistic and controllable video bokeh. These limitations highlight the need for a dedicated framework for temporally coherent, depth-aware video bokeh.

To overcome these limitations, we introduce a new framework that explicitly models scene geometry and leverages strong pre-trained priors to achieve temporally coherent, controllable video bokeh. A central challenge in video bokeh is ensuring smooth blur transitions across depth, particularly near object boundaries, where inaccurate depth often causes artifacts. To address this, we adopt a multi-plane image (MPI) representation, which provides a compact yet explicit encoding of scene geometry. By constructing MPI layers with a progressively widening disparity sampling function, our method captures fine details for in-focus regions while allocating coarser representation to out-of-focus areas, thereby enabling accurate and stable blur transitions across depth boundaries.

Building on this representation, we condition a one-step video diffusion model on MPI layers, guiding the network to synthesize depth-aware bokeh effects that align with subject contours even in cluttered scenes. Unlike prior methods trained from scratch (Luo et al., 2024; Zhang et al., 2019), our framework leverages large-scale pre-trained video diffusion models (Blattmann et al., 2023), whose strong 3D priors provide robust generalization across diverse scenarios and improve structural consistency. To further enhance temporal stability and detail preservation, we design a three-stage progressive training strategy that (i) establishes accurate geometric guidance via MPI-based spatial and temporal modules, (ii) improves robustness and reduces flickering through extended temporal windows with data perturbations, and (iii) refines subject details using a VAE-based enhancement module. Together, these components form a unified framework that directly addresses the core challenges of temporal coherence, geometric accuracy, and visual quality in video bokeh generation.

Extensive experiments on both synthetic and real-world benchmarks demonstrate that our framework produces high-quality, controllable video bokeh and significantly outperforms prior baselines. As illustrated in Fig. 1, our method supports rendering from arbitrary video inputs and allows users to customize the focal plane and bokeh intensity. This capability opens up new possibilities for content creation, cinematic editing, and mobile post-processing. Our main contributions are as follows:

- We propose the first one-step diffusion framework for controllable video bokeh, introducing an MPI-guided conditioning mechanism that injects explicit scene geometry to enable spatially accurate, depth-aware blur synthesis.
- We design a progressive training strategy that improves temporal stability and detail fidelity, yielding realistic and controllable video bokeh across diverse scenarios.
- Our approach achieves state-of-the-art in-focus boundary fidelity and temporal consistency, producing controllable video bokeh for arbitrary input videos with explicit control over the focus plane and bokeh intensity.

## 2 RELATED WORK

### 2.1 CAMERA SIMULATION DIFFUSION MODELS

**Reference Guidance Models**. Recent advancements in camera simulation have demonstrated the potential of leveraging reference videos to guide the motion and behavior of virtual cameras. A line of work (Zhao et al., 2024; Guo et al., 2024) encodes motion cues from reference videos via LoRA (Hu et al., 2022), enabling the diffusion model to replicate specific camera behaviors observed in the

training set. In contrast, MotionClone (Ling et al., 2025) proposes a training-free approach, directly extracting motion patterns from reference videos via spatial and temporal attention modules. Other approaches (Teng et al., 2023; Shi et al., 2024; Wu et al., 2024; Yin et al., 2023) require users to draw reference points to guide lens adjustments, offering more direct control over camera transformations.

**Optical Effects with Diffusion.** While earlier diffusion-based camera simulation methods mainly focused on motion modeling, recent studies have expanded to simulating optical effects such as bokeh. For example, BokehDiffusion (Fortes et al., 2025) and Generative Photography (Yuan et al., 2025) generate high-quality bokeh effects from 2D images, capturing realistic defocus effects. These methods demonstrate the potential of diffusion models for simulating non-geometric camera behavior, but they are restricted to static images and lack temporal modeling, which limits their applicability to video tasks. In contrast, our approach integrates optical effect simulation with pre-trained priors, enabling temporally consistent and controllable bokeh effects for arbitrary video inputs. This work, therefore, bridges the gap between image-based bokeh rendering and the video domain.

## 2.2 Computational Bokeh

**Training-Free Methods**. Early computational bokeh methods are training-free and rely on physically-based or image-based heuristics. Ray tracing (Pharr et al., 2023; Potmesil & Chakravarty, 1981; Akenine-Moller et al., 2019) produces realistic defocus effects, but requires full 3D geometry and is computationally expensive, which limits its practical applicability. Depth-based methods (Wadhwa et al., 2018; Yang et al., 2016) apply scattering or gathering kernels to create spatially varying blur using estimated depth. Matting-based methods (Shen et al., 2016a;b) blur the background based on foreground masks. Although these methods do not require training, they often suffer from artifacts due to inaccurate depth estimation or segmentation errors, leading to unnatural blur transitions.

**Learning-based Methods**. More recent approaches adopt machine learning to improve bokeh synthesis. For example, BokehMe (Peng et al., 2022a) refines depth-based blur using classical methods (Wadhwa et al., 2018) along with neural networks, while models based on Multiplane Images (MPI)(Zhou et al., 2018)(Peng et al., 2022b; Busam et al., 2019; Sheng et al., 2024) decompose scenes into layered representations to render depth-aware effects. Other works explore adaptive kernels or light field approximations (Srinivasan et al., 2018; Kaneko, 2021) to enhance bokeh effects. DeepLens (Lijun et al., 2018) introduces a depth estimation network using depth estimation and foreground segmentation data to enhance the perception of foreground edges End-to-end models trained on paired all-in-focus and bokeh datasets (Ignatov et al., 2020; Seizinger et al., 2025; Dutta et al., 2021) show promise but are limited by dataset biases and fixed parameter simulation. They generally lack the flexibility to control focal planes or simulate custom bokeh intensities.

While image-based bokeh synthesis is well-studied, extending it to videos is challenging. Naive frame-wise methods often introduce temporal flickering and inconsistency. Our work addresses this by a novel MPI-guided conditioning mechanism and leveraging pre-trained video diffusion models for consistent spatial and temporal bokeh rendering.

## 3 Method

An overview of our pipeline is shown in Fig. 2(a). Our framework performs video bokeh rendering in a one-step diffusion scheme, guided by MPI priors. The design comprises three components: (i) a focal-plane–adapted MPI representation that provides geometry-aware spatial priors, (ii) a one-step video diffusion backbone conditioned on MPI-derived and user prompts that generates controllable and meticulous video bokeh rendering, and (iii) a progressive training strategy that improves temporal stability and detail fidelity.

## 3.1 Focal-Plane–Adapted MPI Representation

A central challenge in video bokeh is achieving realistic blur transitions, especially near object boundaries, where depth discontinuities often cause visible artifacts. Prior works (Peng et al., 2022b; Busam et al., 2019) discretize the scene into front-to-back layers with fixed depth values. While effective for simple rendering, such fixed discretizations are poorly aligned with the optics of bokeh: blur radius varies nonlinearly with respect to the focal plane, and equal-depth intervals fail to capture

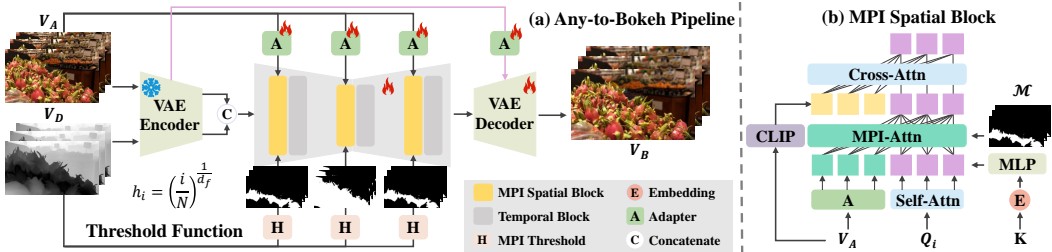

Figure 2: Two key components of Any-to-Bokeh. (a) One-step video bokeh pipeline: receives input of any video and disparity relative to the focal plane to perform the bokeh effect. (b) MPI spatial block: uses the MPI mask $\mathcal{M}$ to prompt MPI spatial block to guide bokeh rendering. The user-defined blur strength $K$ is injected through embedding.

these variations. This mismatch often produces inaccurate blur transitions and artifacts around fine structures. Therefore, we introduce a focal-plane-adapted MPI representation that explicitly drives the geometry prior relevant to bokeh formation. This adaptive scheme yields more accurate depth-dependent blur and ensures smooth transitions across subject contours. Specifically, the optics of the circle of confusion (CoC) relates the blur radius $r$ to disparity:

$$r = K \left| \frac{1}{z} - \frac{1}{z_f} \right| = K|d - d_f|, \tag{1}$$

where $K$ is the user-controlled blur strength; $z$ and $d$ denote depth and disparity, respectively; and $z_f$ and $d_f$ are the corresponding focal-plane values. Since $r$ changes more rapidly near shallow focal planes and more slowly in distant regions, we sample disparity more finely near $d_f$ and more coarsely farther away. Concretely, for each video frame, we predict a disparity map $d$ using a pre-trained depth estimator and define thresholds with a threshold function:

$$h_i = \left( \frac{i}{N} \right)^{\frac{1}{d_f}}, \quad i = 1, 2, \ldots, N - 1, \tag{2}$$

where $1/d_f \in (0, 1]$ acts as a factor, yielding finer sampling for shallow focus. Pixels are assigned to MPI layers using these thresholds, producing a focal-plane–adapted MPI mask

$$\mathcal{M} = \{ m_i \mid |d(m_i) - d_f| < h_i \}, \tag{3}$$

which highlights regions close to the focus and provides finer granularity around depth discontinuities. In addition, we compute a normalized disparity-difference map $V_D$ from $d$ and $d_f$ (an implicit CoC proxy per Eq. (1)). Together, $\mathcal{M}$, $V_D$, and $K$ constitute the geometry prior and user-guided controls that will condition the generator. Unlike fixed front-to-back discretizations (Peng et al., 2022b; Busam et al., 2019), our MPI is defined *relative to the focal plane*, enabling focus-aware sampling and accurate boundary transitions.

## 3.2 One-Step Video Bokeh Diffusion on MPI Priors

To avoid the temporal instability across frames caused by repeated iterations of the multi-step diffusion model and the high computational cost of multi-step inference, we formulate the generation of video bokeh as a one-step diffusion guided by the MPI prior. Building on Stable Video Diffusion (SVD) (Blattmann et al., 2023), we remove stochastic sampling and adopt a one-step U-Net that directly predicts the output frames.

The backbone is conditioned on three explicit signals: i) **normalized disparity difference** $V_D$ (focus proximity); ii) **blur strength** $K$ (bokeh intensity); iii) **focal-plane–adapted MPI mask** $\mathcal{M}$ (geometry prior). This conditioning provides precise control over focus placement and bokeh intensity, while promoting frame-to-frame consistency.

## 3.3 MPI Spatial Blocks

To incorporate geometry into feature processing, we introduce MPI Attention, a gated attention mechanism inspired by (Li et al., 2023), integrated into the U-Net's spatial blocks, referred to in

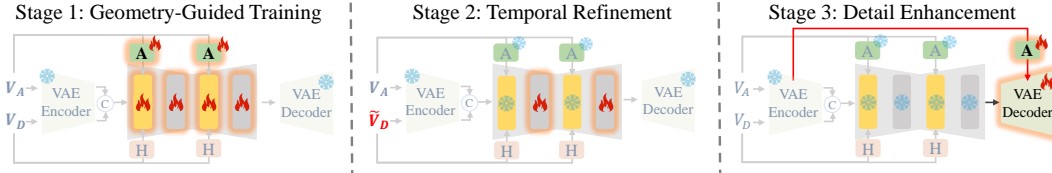

Figure 3: **Progressive Training Strategy**: Stage 1: Train the whole U-Net and adapters. Stage 2: Refine the temporal block with the disturbance. Stage 3: Fine-tuning VAE decoder. We desaturated the colors in the same areas.

Fig. 2(b) as MPI spatial blocks. Let $\mathbf{Q} = \{\mathbf{Q}_1, \ldots, \mathbf{Q}_i\}$ be query tokens from the current block, $\mathbf{V}_A$ be visual tokens from the input video, and $\gamma$ a learnable gate (initialized to zero). We modulate queries with $\mathbf{K}$ and guide attention with the focal-plane–adapted MPI mask:

$$\hat{\mathbf{Q}} = \mathbf{Q} + \tanh(\gamma) \cdot \mathrm{TS}\left(\mathrm{Attn}\left([\mathbf{Q} + \Phi_M(E(\mathbf{K})), \Phi_A(\mathbf{V}_A)], \bar{\mathcal{M}}\right)\right), \qquad (4)$$

where $E(\mathbf{K})$ is a Fourier embedding (Mildenhall et al., 2021) of $\mathbf{K}$, $\Phi_M(\cdot)$ and $\Phi_A(\cdot)$ are lightweight MLPs, $\mathrm{TS}(\cdot)$ is a token-selection operator that selects the outputs corresponding to the tokens of $\mathbf{Q}$ after the attention operation, discarding the auxiliary tokens $\mathbf{V}_A$; $[\cdot, \cdot]$ denotes concatenation, and $\bar{\mathcal{M}} = [\mathbf{1}, \mathcal{M}]$ denotes the mask augmented by padding an all-ones matrix, which is used in the masked-attention computation. We inject *near-focus* masks in shallow U-Net blocks to refine local transitions and *wider-interval* masks in deeper U-Net blocks for global context; mask resolutions are aligned to each block by bilinear interpolation. This design steers attention toward focus-relevant regions while maintaining global structure, enabling geometry consistent, contour-aligned bokeh.

## 3.4 PROGRESSIVE TRAINING STRATEGY

Although the one-step video diffusion model conditioned on MPI priors provides strong controllability, training such a model directly is challenging. In practice, three issues must be addressed: (i) maintaining temporal consistency across frames, (ii) ensuring robustness against imperfect depth estimation, and (iii) preserving high-frequency details that are often lost during encoding. To systematically tackle these challenges, we adopt a three-stage progressive training strategy, as illustrated in Fig. 3.

**Stage 1: Geometry-Guided Training.** We begin by fine-tuning the MPI spatial block, temporal block, and adapters on clean data. Training without stochastic noise encourages the network to fully exploit the focal-plane-adapted MPI mask, enabling it to learn spatially accurate, depth-aware blur effects. This stage allows the model to establish a reliable foundation for spatial geometry and to build initial temporal consistency across video frames.

**Stage 2: Temporal Refinement with Disparity Noise.** A key difficulty in video bokeh is robustness to imperfect or noisy depth estimation, especially around focal boundaries. To address this, we freeze the spatial MPI block and train only the temporal modules on longer sequences while injecting targeted disparity perturbations. Specifically, we apply elastic transforms (Buslaev et al., 2020) to distort depth locally, add Perlin noise (Perlin, 1985) to simulate natural inconsistencies, and use morphological operations to exaggerate boundary transitions. This perturbation setting prevents the spatial modules from being overwhelmed while teaching the temporal block to tolerate depth noise and reduce flickering. As a result, the model develops longer temporal memory and stronger resilience to real-world variations.

**Stage 3: Detail Enhancement via VAE Decoder.** Even with stable geometric and temporal modeling, fine details may be lost due to VAE-based compression. In the final stage, we fine-tune the VAE decoder and its adapter using clean data, introducing a skip connection from the encoder to reuse spatially rich features (Parmar et al., 2024). To emphasize texture fidelity, we combine an image-space $L_1$ loss with a gradient-based texture loss $\mathcal{L}_t$:

$$\mathcal{L}_t = \sum_{x,y} \left[ \left( \nabla_x \hat{\mathbf{V}}_B(x, y) - \nabla_x \mathbf{V}_B(x, y) \right)^2 + \left( \nabla_y \hat{\mathbf{V}}_B(x, y) - \nabla_y \mathbf{V}_B(x, y) \right)^2 \right], \qquad (5)$$

where $\hat{\mathbf{V}}_B$ and $\mathbf{V}_B$ denote the predicted and ground truth frames. This loss encourages sharper edges and more realistic textures, thereby enhancing the perceptual quality of the bokeh effect.

Through this progressive strategy, the model gradually learns spatial geometry, temporal robustness, and detail recovery, yielding temporally coherent, depth-aware, and visually realistic video bokeh.

### 3.5 WEIGHTED OVERLAP INFERENCE STRATEGY

Naively splitting a video into overlapping segments and processing each segment independently can result in noticeable temporal discontinuities at the segment boundaries. These discontinuities often manifest as flickering, misaligned blur transitions, or mismatched bokeh effects, disrupting the smoothness and continuity of the video. To address this, we propose a weighted overlapping inference strategy (WOIS). Specifically, we divide the input video into $P$ overlapping segments, each containing $2L$ frames, with adjacent segments overlapping by exactly $L$ frames. For the j-th frame in the overlap region, denoted as $\hat{V}_B^i[j]$, we compute the final output by combining adjacent frames from overlapping segments using a weighted average:

$$\tilde{\boldsymbol{V}}_B^i[j] = \gamma_j\,\hat{\boldsymbol{V}}_B^i[j] + (1 - \gamma_j)\,\hat{\boldsymbol{V}}_B^{i+1}[j + L], \quad i \in \{1, 2, \ldots, P\}, \tag{6}$$

where the weighting factor $\gamma_j$ is defined as:

$$\gamma_j = \frac{1}{2}\left(1 + \cos(\frac{\pi j}{L})\right), \tag{7}$$

which smoothly blends the frames at the boundaries, ensuring a gradual transition between segments. The cosine-based weighting function assigns higher weights to the central frames of each segment, reducing the influence of the boundary frames and minimizing visual artifacts at the segment edges. This strategy enables our framework to process videos of arbitrary length, providing a robust solution to the problem of temporal flickering and boundary artifacts in video bokeh generation.

## 4 EXPERIMENTS

### 4.1 IMPLEMENTATION DETAILS

**Dataset**. Currently, paired datasets for all-in-focus and bokeh videos are lacking. Existing computational bokeh datasets (Ignatov et al., 2020; Seizinger et al., 2025; Dutta et al., 2021) primarily consist of image pairs but lack temporal consistency, and previous video-based works (Luo et al., 2024) often omit foreground motion. To address this, we build upon prior work (Peng et al., 2025) and adopt a synthetic approach to generate paired all-in-focus and bokeh video sequences. For accurate foreground extraction, we use objects from the video matting dataset (Lin et al., 2021), isolating them using the alpha channel for precise segmentation. We also incorporate the image matting dataset (Li et al., 2022; Hu et al., 2024) and collect 1,300 background images from the intern dataset and background dataset (Lin et al., 2021). In each video, we randomly select background and foreground clips, simulating real-world camera adjustments such as focal plane and aperture changes. The foreground objects are moved along random 3D trajectories, each containing 25 frames. All videos in the dataset have a resolution of $1024 \times 576$ pixels. Using a ray-tracing-based method (Peng et al., 2022b), we generate accurate bokeh effects, ensuring temporal coherence across frames.

**Training and Inference**. In this work, we use SVD (Blattmann et al., 2023) as our base model. During training, the process is divided into three stages, and the value of $N$ in Eq. (2) is set to 4. Stage 1: We train using 4-frame video sequences with the Adam optimizer at a learning rate of 1e-5. Stage 2: We train with 8-frame video sequences, using a learning rate of 5e-6 and a depth perturbation probability of 0.5. Stage 3: We fine-tune the VAE decoder using the same learning rate as in Stage 2. For all stages, the video resolution is set to $1024 \times 576$ with a batch size of 1, and training is performed across 4 Nvidia H800 GPUs. During inference, we use the weighted overlap inference strategy where long videos are divided into clips of 8 frames, with each clip having a 4-frame overlap.

**Test Setting**. We validate our model on both synthetic and real-world datasets. First, we synthesize a test set of 200 videos with varying bokeh strength and focus planes. We report the following metrics: LPIPS (Zhang et al., 2018), PSNR and SSIM (Wang et al., 2004) for image fidelity; VFID (Wang et al., 2018) with I3D features (Carreira & Zisserman, 2017) (denoted VFID-I) and FVD (Ge et al., 2024) for video quality; and the relation metric (Luo et al., 2024) (denoted RM) for temporal consistency, which computes pixel-wise differences between adjacent frames. Additionally, we calculate flow

Table 1: Quantitative comparison of Any-to-Bokeh. The best metric scores in each column are marked in **bold** for clarity."↓" or "↑" indicate lower or higher values are better.

| Method | FD↓ | RM↓ | VFID-I↓ | FVD↓ | SSIM↑ | PSNR↑ | LPIPS↓ | VEPI↑ | Time↓ |
|--------|-----|-----|---------|------|-------|-------|--------|-------|-------|
| DeepLens | 1.162 | 0.030 | 16.042 | 125.338 | 0.819 | 24.574 | 0.183 | 0.715 | 0.226 |
| BokehDiff | 0.660 | 0.021 | 7.395 | 65.678 | 0.834 | 27.525 | 0.127 | 0.859 | 0.799 |
| BokehMe | 0.536 | 0.013 | 8.633 | 39.102 | 0.936 | 27.992 | 0.060 | 0.937 | **0.103** |
| Dr.Bokeh | 0.522 | 0.011 | 6.097 | 32.710 | 0.950 | 31.273 | 0.046 | 0.863 | 2.729 |
| MPIB | 0.481 | 0.011 | 5.444 | 35.766 | 0.950 | 31.390 | 0.040 | 0.921 | 0.521 |
| **Ours** | **0.431** | **0.007** | **1.479** | **9.005** | **0.974** | **38.899** | 0.019 | **0.944** | 0.363 |

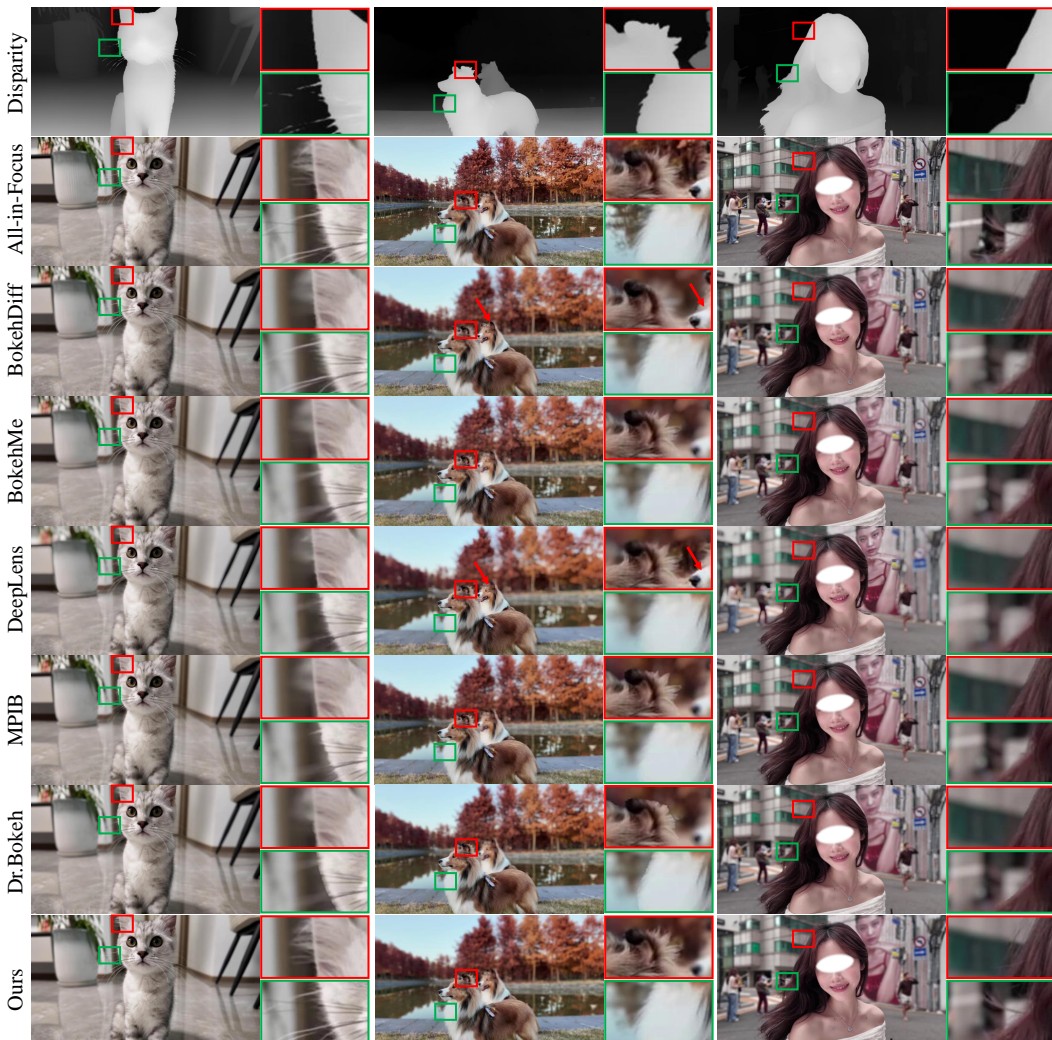

Figure 4: Qualitative Results on Real-World Video frames. To highlight the differences, we zoom in on the red and green regions. Red arrows indicate incorrectly focused areas.

difference (FD) between predicted and ground truth frames using RAFT (Teed & Deng, 2020). For real-world scenarios, we design VEPI, inspired by prior work (Joseph et al., 2017), to assess the model's ability to preserve detail at the edges of the focused subject. We apply the VEPI metric to the DAVIS dataset (Perazzi et al., 2016), with implementation details described in Appendix A.4. We evaluate 25-frame videos and compute the mean inference time.

## 4.2 RESULTS ON TEST DATASET

To verify the performance of our proposed method, we compare it with four existing computational bokeh methods: DeepLens (Lijun et al., 2018), BokehMe (Peng et al., 2022a), Dr.Bokeh (Sheng et al.,

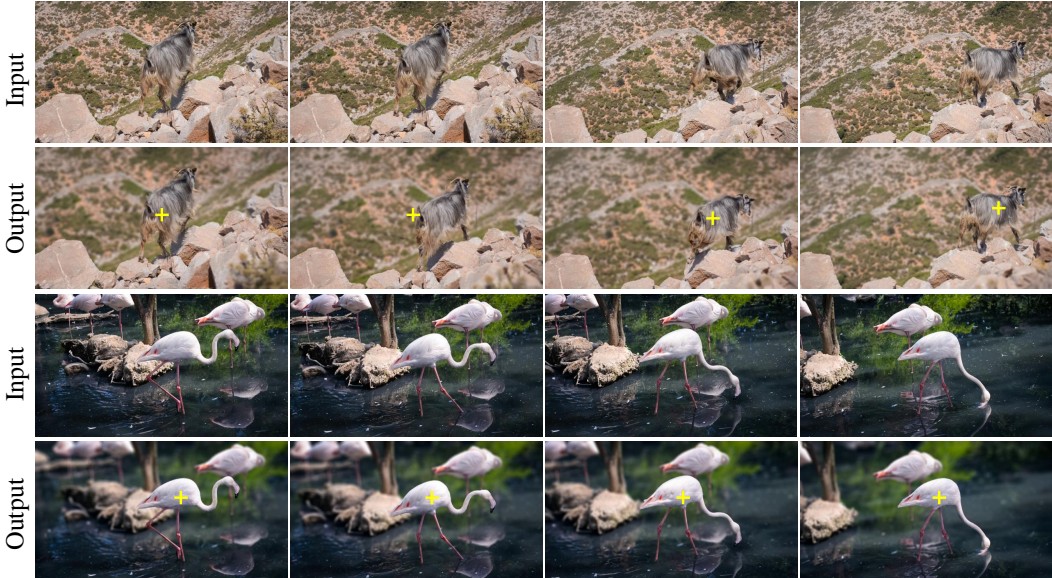

Figure 5: Visualization of generated bokeh effects on DAVIS dataset. The yellow cross represents the focus subject. Please zoom in to see the details.

2024), MPIB (Peng et al., 2022b), and BokehDiff (Zhu et al., 2025). Specifically, to compare with traditional MPI-based bokeh methods, MPIB and Dr.Bokeh use the conventional approach, which discretize the scene into front-to-back layers with a fixed depth value. Only one prior work (Luo et al., 2024) addresses video bokeh, but it is not publicly available for comparison.

**Quantitative Results**. As shown in Tab. 1, our method consistently outperforms all other approaches across all metrics. Specifically, Any-to-Bokeh achieves the lowest FD and the best RM, demonstrating superior temporal consistency. Furthermore, Any-to-Bokeh surpasses the baselines in video quality, attaining the lowest VFID-I and FVD, along with the highest SSIM and PSNR, demonstrating its capability to produce high-quality image fidelity and video bokeh. In real-world scenarios, our approach achieves the highest VEPI, demonstrating better blur transitions. Furthermore, compared to traditional MPI methods, our approach exhibits an efficiency advantage in inference time.

**Qualitative Results**. To further highlight the visual advantages of our approach, we present three examples in Fig. 4. Any-to-Bokeh is able to preserve detailed foreground textures, such as animal fur and flowing hair, even under poor depth prediction conditions. Additionally, it produces bokeh effects that adhere to optical principles across different depth planes, such as the cat's tail in the first column and the dog in the second column at a farther distance. Traditional MPI-based methods, such as MPIB (Peng et al., 2022b) and Dr.Bokeh (Sheng et al., 2024), struggle to blend different depth planes naturally, resulting in color bleeding, as seen in the dog's ear in the second column. BokehDiff (Zhu et al., 2025) and Deeplens (Lijun et al., 2018) can preserve some foreground details, but produce bokeh results that conflict with depth, such as the dog in the second column marked with a red arrow. Moreover, BokehDiff consistently exhibits loss of fine details in its results. BokehMe (Peng et al., 2022a) heavily relies on accurate depth information and fails to maintain fur details, as seen in the in-focus subject in the first and third columns when depth is inaccurate. Additional examples are provided in Fig. 10, Fig. 11, Fig. 12, and Fig. 13.

**Quantitative Comparison Under Inaccurate Disparity**. As illustrated in Tab. 2, we introduce three types of perturbations to the disparity maps. As shown in Fig. 6, we report three metrics across three perturbation modes (II–IV). As the disparity perturbation increases from mode II to IV, our method maintains almost unchanged PSNR and consistently outperforms all baselines, demonstrating stable reconstruction capability under complex conditions, whereas competing methods exhibit varying degrees of degradation. In terms of video quality, our method maintains the lowest FVD across all perturbation modes, while other methods exhibit notable fluctuations or degradation (e.g., BokehDiff rises from 65 to 80, and MPIB shows a sharp increase at mode IV). Regarding temporal consistency, our method consistently achieves the lowest FD and exhibits only minimal fluctuations even under severe disparity perturbations, whereas competing methods display an overall upward trend.

Table 2: Mapping of Perturbation Modes.

| Perturbation Mode | Gaussian Blur | Elastic Transform | Erosion Dilation |
|:---:|:---:|:---:|:---:|
| I | | | |
| II | ✓ | | |
| III | ✓ | ✓ | |
| IV | ✓ | ✓ | ✓ |

Table 3: Results on human preference.

| Baseline | Preference |
|:---:|:---:|
| Ours vs. DeepLens | 96.9% / 3.1% |
| Ours vs. BokehMe | 77.1% / 22.9% |
| Ours vs. MPIB | 62.9% / 37.1% |
| Ours vs. Dr.Bokeh | 77.8% / 22.2% |
| Ours vs. BokehDiff | 75.7% / 24.3% |

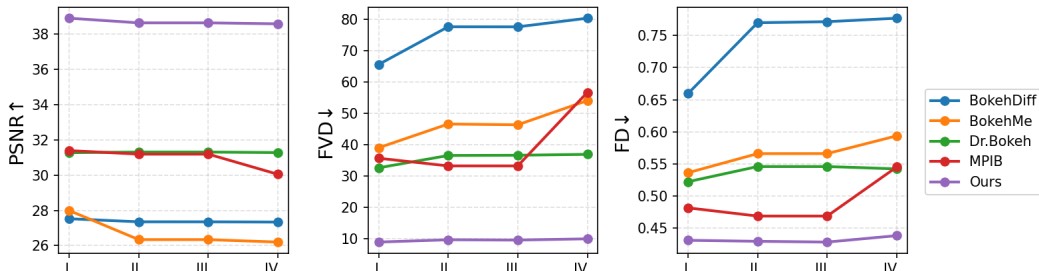

Figure 6: Evaluation on the synthetic test set with inaccurate disparity maps. The horizontal axis corresponds to the perturbation modes defined in Tab. 2.

**Computational Cost Comparison.** We provide a comparison of computational cost in Tab. 4. All methods are evaluated on single-frame inputs at a resolution of 576×1024 on the H800 GPU to ensure fairness. For BokehMe and Dr.Bokeh, which rely on custom CUDA-based renderers, full FLOPs and parameter counts are not directly comparable.

Table 4: Results of the computational cost comparison.

| Method | Time (s) | Params (M) | GFLOPs | VRAM (GB) |
|:---|:---:|:---:|:---:|:---:|
| BokehMe | 0.103 | *1.41 | *169.71 | 1.4 |
| MPIB | 0.521 | 27.46 | 1418.50 | 3.3 |
| Dr.Bokeh | 2.729 | — | — | 2.7 |
| BokehDiff | 0.799 | 2459 | 3430 | 18.4 |
| Ours | 0.094 | 1880 | 3620 | 13.6 |

Therefore, for BokehMe, we report only the neural network component (marked with "*"). Despite using a larger backbone, our model achieves the shortest runtime among all methods. This efficiency stems from our end-to-end optimization and a highly parallelizable architecture. Compared with the diffusion-based method BokehDiff, our approach is more efficient in terms of parameter count and VRAM usage, and it also offers a clear advantage in inference time over the traditional MPI-based method MPIB and Dr.Bokeh.

**User Study on Real-world Videos.** We test on the DAVIS (Perazzi et al., 2016) dataset. As shown in Fig. 5, our method generates bokeh effects that follow optical principles and maintain strong temporal consistency. We conducted a user study on them to better evaluate different methods from a subjective perspective. We randomly selected 20 videos from the DAVIS dataset and rendered them using various methods. Each participant viewed two videos at a time: one generated by our method and one from a randomly selected baseline. The videos were presented in random order. Participants were asked to choose the method that produced the most aesthetically pleasing bokeh effect based on their personal preference. If they found it difficult to decide, they were allowed to skip making a selection. The user study involved 51 participants and provided 1020 ratings across these video sets. The results in Tab. 3 indicate that our method was preferred over the others, demonstrating a higher human preference for the bokeh effects generated by our approach.

### 4.3 ABLATION STUDIES

We analyze the impact of each component in the Any-to-Bokeh framework on the synthetic test dataset (results in Tab. 5), evaluate the VAE's contribution to video detail quality (as shown in Tab. 6), and assess the effect of temporal refinement on model robustness (as shown in Tab. 7).

**MPI Spatial Block Ablation.** Lines 3 and 4 in Tab. 5 show the ablation results for the MPI block. First, we remove the MPI module and use the original attention block, adding the blur strength ($K$) embedding directly to the original SVD embedding. From the VFID and FVD metrics, we observe that our model benefits significantly from the MPI blocks, leading to better video quality. Additionally, the MPI block enhances temporal consistency and single-frame quality, which are

Table 5: Ablation study of Any-to-Bokeh module. "MPI": MPI spatial block. "OS": one-step inference schedule. "WOIS": weighted overlap inference strategy. "TR": temporal refinement.

| Variants | MPI | OS | WOIS | TR | FD↓ | RM↓ | VFID-I↓ | FVD↓ | SSIM↑ | PSNR↑ | LPIPS↓ |
|---|---|---|---|---|---|---|---|---|---|---|---|
| #1 | ✓ | ✓ | ✓ | ✓ | 0.517 | 0.013 | 3.865 | 18.922 | 0.907 | 32.250 | 0.051 |
| #2 | ✓ | ✓ | ✓ | - | 0.540 | 0.013 | 4.209 | 20.743 | 0.905 | 32.035 | 0.052 |
| #3 | ✓ | ✓ | - | - | 0.551 | 0.013 | 4.521 | 21.941 | 0.905 | 31.936 | 0.054 |
| #4 | - | ✓ | - | - | 0.573 | 0.013 | 4.930 | 23.828 | 0.903 | 31.551 | 0.059 |
| #5 | ✓ | - | - | - | 0.791 | 0.014 | 8.912 | 68.910 | 0.878 | 29.309 | 0.069 |

Table 6: Ablation study on FV.

| FV | FVD↓ | SSIM↑ | PSNR↑ | LPIPS↓ |
|---|---|---|---|---|
| ✓ | 9.005 | 0.974 | 38.899 | 0.019 |
| - | 18.922 | 0.907 | 32.250 | 0.051 |

Table 7: Ablation study on TR under noisy depth.

| TR | FD↓ | VFID-I↓ | FVD↓ | SSIM↑ | PSNR↑ | LPIPS↓ |
|---|---|---|---|---|---|---|
| ✓ | 0.531 | 3.880 | 19.648 | 0.906 | 32.182 | 0.050 |
| - | 0.566 | 4.442 | 22.452 | 0.904 | 31.890 | 0.054 |

crucial for the bokeh rendering task. Additionally, we report results for two simple alternatives to MPI Attention, further demonstrating its effectiveness. Please refer to Appendix A.5.1.

**Effectiveness of One-Step Inference**. We add noise to the input video features and retrain a multi-step denoising model, comparing its performance with the one-step inference schedule. As shown in the 3rd and 5th rows of Tab. 5, the one-step inference schedule substantially improves all metrics, highlighting its ability to achieve more temporally consistent and higher-quality bokeh effects.

**Effectiveness of Weighted Overlap Inference Strategy (WOIS)**. By incorporating WOIS, our model achieves improved temporal consistency, with the flow difference (FD) decreasing from 0.551 to 0.540. Additionally, higher FVD and VFID scores indicate enhanced video quality. Furthermore, the PSNR also shows an increase, reflecting improved single-frame fidelity. More detail are provided in Appendix A.5.2.

**Effectiveness of Progressive Training Strategy**. We begin the ablation study by evaluating the full model. In the second row in Tab. 5, we remove the temporal block refinement (TR) from stage 2, resulting in a decrease in temporal consistency (FD: 0.517 vs. 0.540). This highlights the importance of temporal block refinement in maintaining temporal coherence across video frames. As shown in Tab. 6, we alleviated the loss of high-frequency information by fine-tuning the VAE (FV), which effectively improves both single-frame consistency and overall video quality.

**Ablation Study on Robustness.** To test the contribution of TR to robustness, we introduce perturbations to the disparity in the test dataset using elastic transform (Buslaev et al., 2020), Gaussian blur, and morphological transformations. As shown in the last two rows in Tab. 7, TR leads to improvements across all metrics, with particularly noticeable gains in temporal consistency (FD) and video quality (VFID-I and FVD). These results demonstrate that training the temporal blocks with noisy data during the TR stage effectively enhances the model's robustness.

## 5  CONCLUSIONS

In this work, we propose the first one-step diffusion framework for controllable video bokeh. By incorporating an MPI-guided conditioning mechanism that injects explicit scene geometry, our method achieves higher-quality and more temporally consistent bokeh effects. Additionally, we introduce a progressive training strategy that enhances robustness and detail preservation, significantly improving bokeh quality. We hope our findings inspire further exploration of optical phenomena in editing models, driving advancements in their application to content creation and visual effects. By better understanding these phenomena, we aim to enhance the realism and flexibility of future editing models, enabling more creative possibilities for the industry.

**Limitations**: While Any-to-Bokeh achieves significant improvements over existing methods, it still has two main limitations. First, due to limited computational resources, we did not investigate training with larger batch sizes. With more GPUs and a larger-scale training setup, the model could further improve its ability to handle complex scenes. Second, for long videos, we cannot process the entire sequence in a single forward pass; instead, inference relies on repeatedly applying the WOIS strategy over overlapping segments, which sacrifices some efficiency.

ETHICS STATEMENT

Our curated dataset is derived from publicly available matting datasets and an internal dataset. We use these assets in accordance with their original licenses and terms of use. No new data involving human subjects were collected, and all visualizations preserve privacy. We confirm that our method and experiments do not raise additional ethical concerns.

REPRODUCIBILITY STATEMENT

We are committed to ensuring the reproducibility of our work. To this end, we provide detailed descriptions of our methodology, data, and experimental setup throughout the paper and appendices. Following the blind review period, we plan to release the code for Any-to-Bokeh along with our model weights.

ACKNOWLEDGMENTS

This research was supported by The National Nature Science Foundation of China (Grant Nos: 62432014, 62402417, 62273301, 62273302), in part by "Pioneer" and "Leading Goose" R&D Program of Zhejiang (Grant No. 2025C02026), in part by the Key R&D Program of Ningbo (Grant Nos: 2024Z115, 2025Z035), in part by Yongjiang Talent Introduction Programme (Grant No: 2023A-197-G).

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

# A APPENDIX

CONTENTS

In the appendix, we begin with the LLM usage statement, followed by a detailed description of our dataset, user study, and the VEPI metric. We then report additional ablation studies, covering MPI attention and the weighted overlap inference strategy. Finally, we present additional visualization results to further demonstrate the superiority of our approach.

## A.1 USAGE OF LLM

In this study, a large language model (LLM) was used to polish the writing and elevate the overall textual quality.

## A.2 THE DETAILS OF DATASET

As mentioned on Sec. 4.1, we use objects from the video matting dataset (Lin et al., 2021) and image matting dataset (Li et al., 2022; Hu et al., 2024) and collect 1,300 background images from the intern dataset and background dataset (Lin et al., 2021). Following MPIB (Peng et al., 2022b), we use a ray-tracing-based method to generate accurate bokeh effects. Specifically, we assume that the disparities of all images are planar, with their size and position randomly determined. The disparity map $d$ is then set as a plane equation of pixel coordinates $(x, y)$

$$d = \frac{1 - ax - by}{c},\tag{8}$$

where $a, b,$ and $c$ are parameters that define the spatial depth relationship between pixels. For each pixel, we sample multiple rays passing through the lens, find the intersection of each ray with the scene, and project this intersection onto the sensor plane to obtain the final render results. As shown in Fig. 7, for each video, we randomly select background and foreground clips, as well as the focal plane and aperture. The foreground objects are moved along random 3D trajectories, with movement in six dimensions: forward and backward, left and right, up and down. Each video contains 25 frames, and all videos in the dataset have a resolution of $1024 \times 576$ pixels. To evaluate model performance, we use the same approach to synthesize a test set of 200 videos.

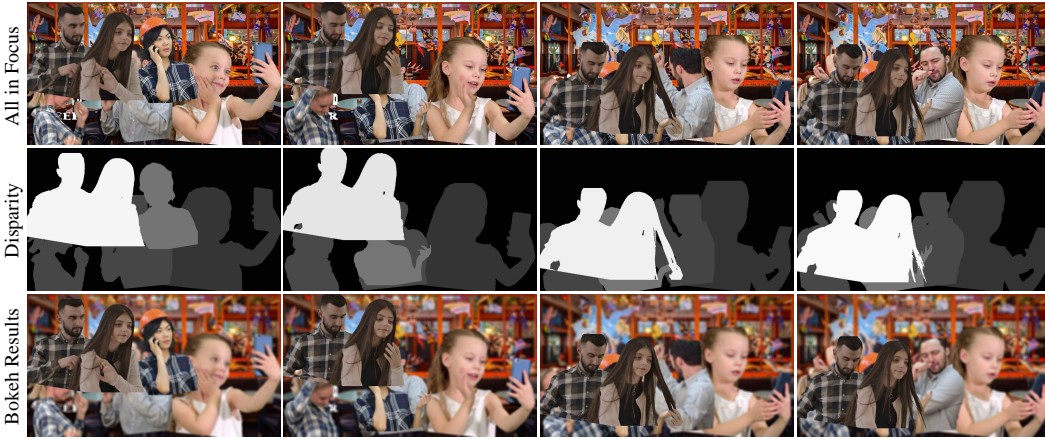

Figure 7: Example of synthetic datasets, we randomly define the focal plane, the position of each foreground, and the blur intensity.

## A.3 USER STUDY DETAILS

Given the subjective nature of perceiving video bokeh rendering results, and the absence of ground truth (GT) data for video bokeh, we conducted a user study on real-world videos to evaluate different methods from a subjective perspective. We randomly selected 20 videos from the DAVIS dataset, each with a resolution of $1024 \times 576$, featuring subjects such as people, animals, and various other objects. Since ground truth disparity maps were unavailable, we generated disparity maps using a depth prediction model (Chen et al., 2025). Different methods were used to render the videos with the same control parameters. During testing, the videos were presented in random order to avoid bias. As shown in the interface (Fig. 8), participants were asked to select the method that produced the

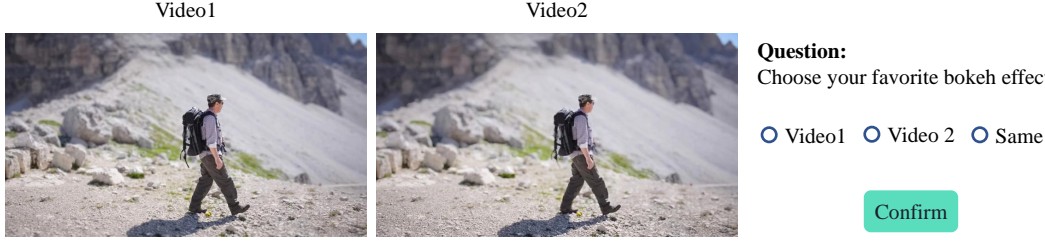

Figure 8: The user study interface, where they were asked to select their preferred videos.

most consistent and aesthetically pleasing bokeh effect. If they found it difficult to decide, they were allowed to skip making a selection.

### A.4 VEPI METRIC DETAILS

We adapt the Edge Preservation Index (EPI) from prior image-based work (Joseph et al., 2017). In VEPI, a Laplacian kernel is applied to generate binary edge maps for both the reference image and the bokeh-rendered image:

$$\text{VEPI} = \frac{\Gamma(\Delta s - \overline{\Delta s}, \Delta \hat{s} - \overline{\Delta \hat{s}})}{\sqrt{\Gamma(\Delta s - \overline{\Delta s}, \Delta s - \overline{\Delta s}) \cdot \Gamma(\Delta \hat{s} - \overline{\Delta \hat{s}}, \Delta \hat{s} - \overline{\Delta \hat{s}})}}, \tag{9}$$

where

$$\Gamma(s_1, s_2) = \sum_{i,j \in \text{ROI}} s_1(i,j) \cdot s_2(i,j). \tag{10}$$

ROI denotes the unblurred foreground region. $\Delta s(i,j)$ and $\Delta \hat{s}(i,j)$ are the Laplacian-filtered versions of the ROI in the reference $s(i,j)$ and the bokeh rendering $\hat{s}(i,j)$, respectively. Since this metric relies on precise foreground delineation, we employ the DAVIS dataset (Perazzi et al., 2016), which provides densely annotated video segmentation. We compute VEPI for all videos in the DAVIS dataset, using the provided segmentation masks as the ROI, and report the mean value across all videos as the final result.

### A.5 MORE ABLATION STUDY

#### A.5.1 MPI ATTENTION

Table 8: Ablation results for MPI attention. "MPI" refers to MPI attention, while "PR" indicates pretrained SVD weights.

| Variants | MPI | PR | FD↓ | RM↓ | VFID-I↓ | FVD↓ | SSIM↑ | PSNR↑ |
|----------|-----|-----|-------|-------|---------|--------|-------|--------|
| #1 | ✓ | ✓ | 0.551 | 0.013 | 4.521 | 21.941 | 0.905 | 31.936 |
| #2 | - | ✓ | 0.568 | 0.014 | 4.714 | 22.556 | 0.901 | 31.575 |
| #3 | ✓ | - | 0.586 | 0.014 | 4.537 | 27.743 | 0.893 | 30.988 |

To validate the role of MPI Attention within the MPI spatial block, we report results for two simple alternatives. First, we discard the guidance of the focal-plane–adapted MPI mask and replace MPI attention with standard self-attention. As shown in the second row of Tab. 8, all metrics decrease, indicating that the focal-plane–adapted MPI mask effectively provides a geometry prior, resulting in temporally consistent and high-quality bokeh. Second, we remove the pretrained SVD weights and train the MPI spatial block from scratch. The results in the third row of Tab. 8 demonstrate that the pretrained 3D priors significantly improve model performance.

#### A.5.2 WEIGHTED OVERLAP INFERENCE STRATEGY

As introduced in Sec. 3.5, the weighted overlap inference strategy employs a weighting factor for blending overlapping frames. We evaluate two schemes: (i) cosine-based blending, which emphasizes

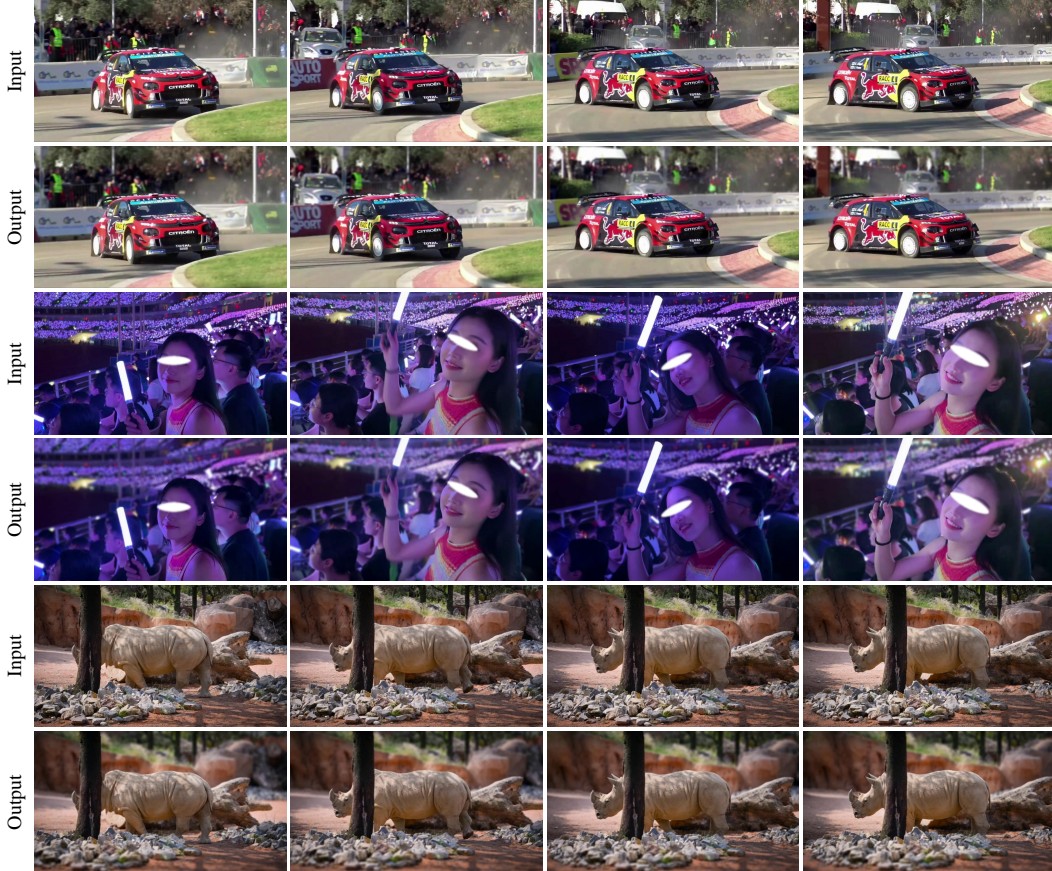

Figure 9: Visualization results on challenging real-world scenes with fast motion, varying lighting, and occlusions.

central frames within each segment, and (ii) linear-based blending, which assigns weights linearly as $\gamma_j = \frac{j}{L}$. As shown in Tab. 9, cosine-based blending provides superior temporal consistency and higher-quality bokeh. Consequently, we adopt the cosine-based blending (Eq. (7)) as the weighting factor for overlapping frame fusion.

Table 9: Results for different blending strategy.

| Method | FD↓ | RM↓ | VFID-I↓ | FVD↓ | SSIM↑ | PSNR↑ |
|---|---|---|---|---|---|---|
| Linear | 0.422 | 0.007 | 1.565 | 9.168 | 0.972 | 38.706 |
| Cosine | 0.431 | 0.007 | 1.479 | 9.005 | 0.974 | 38.899 |

## A.6 MORE VISUALIZATION RESULTS

**More Videos Visualization**: Real-world scenes are often highly complex. To further demonstrate the effectiveness of our method in such settings, we present rendering results on three challenging scenarios characterized by fast motion, varying lighting, and occlusions. As shown in Fig. 9, our method still produces high-quality renderings under these challenging scenarios.

**Visualization Results under Suboptimal Depth Inputs**: We present further comparisons between Any-to-Bokeh and baselines to validate its ability to render bokeh under poor depth conditions. In Fig. 10 and Fig. 11, errors in depth prediction are visible along fine edges, such as hair, dog fur, or earrings. BokehMe, DeepLens, MPIB, and Dr.Bokeh rely heavily on accurate depth; in regions with missing or erroneous depth, they fail to capture foreground edges, resulting in degraded bokeh. Although BokehDiff can render some edge textures, it struggles to remain consistent with the input. Our method effectively addresses these challenges, producing spatially accurate, depth-aware blur.

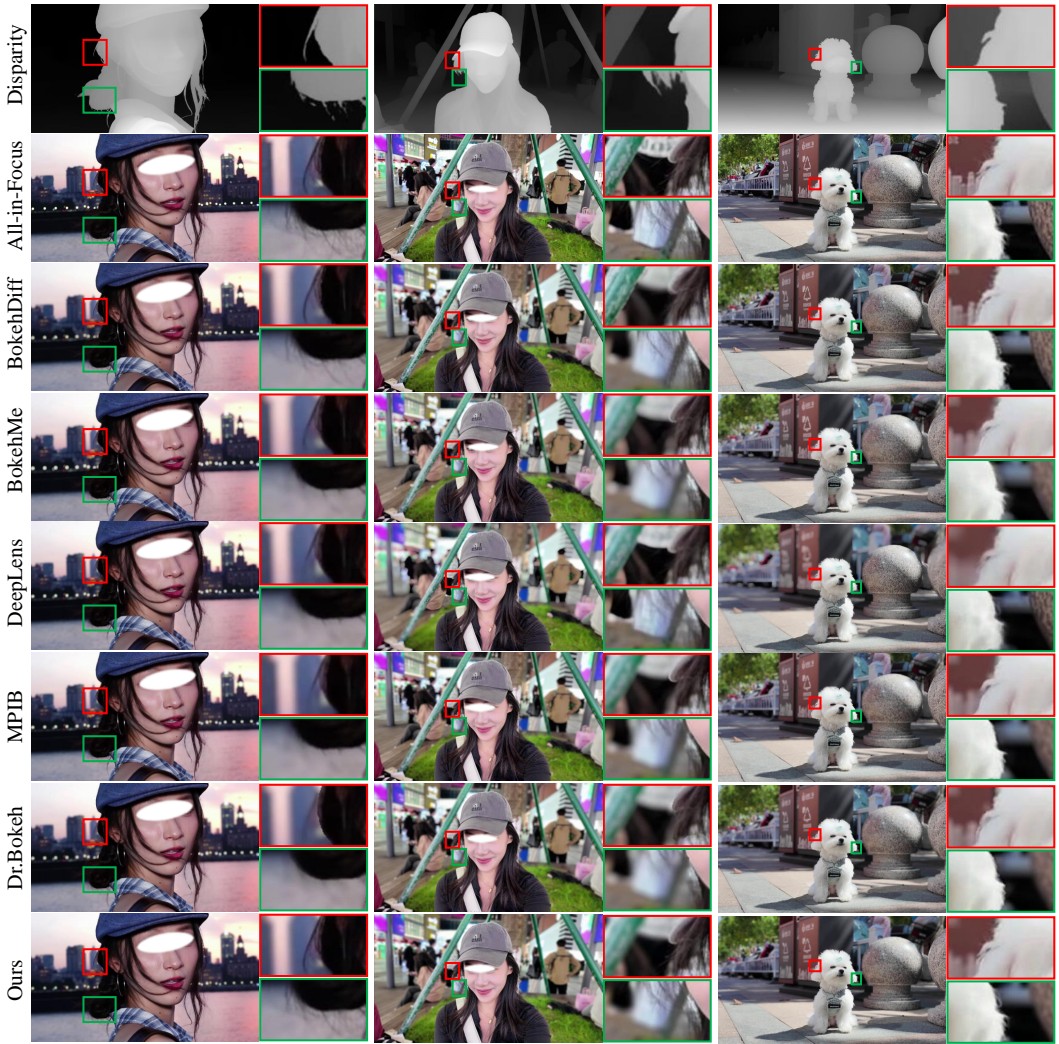

Figure 10: Comparison results with baselines under suboptimal depth inputs. To highlight the differences, we zoom in on the red and green regions.

**Real-World Videos Comparison**: We further compare our model with baselines using two representative examples. As shown in Fig. 12 and Fig. 13, our method preserves the fine details of the in-focus subject, achieving more visually coherent bokeh. Even under challenging depth conditions, it accurately captures the edges of the subject, enhancing the bokeh realism. In contrast, BokehDiff struggles with the accurate rendering of subject textures, such as the characters on the hat (green box) and hair strands (red box). Furthermore, BokehMe, DeepLens, MPIB, and Dr.Bokeh are unable to preserve hair details within the same focal plane due to poor depth estimation.

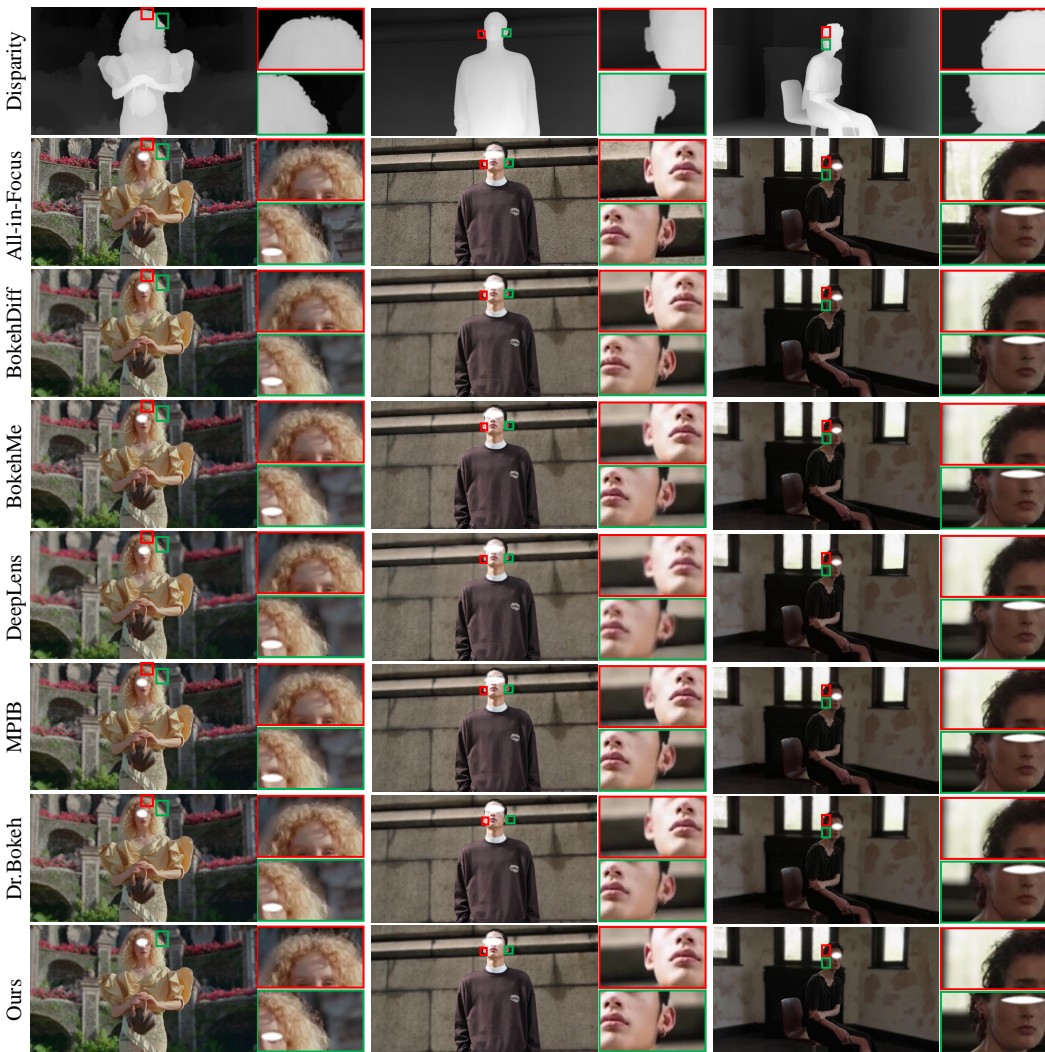

Figure 11: Comparison results with baselines under suboptimal depth inputs. To highlight the differences, we zoom in on the red and green regions.

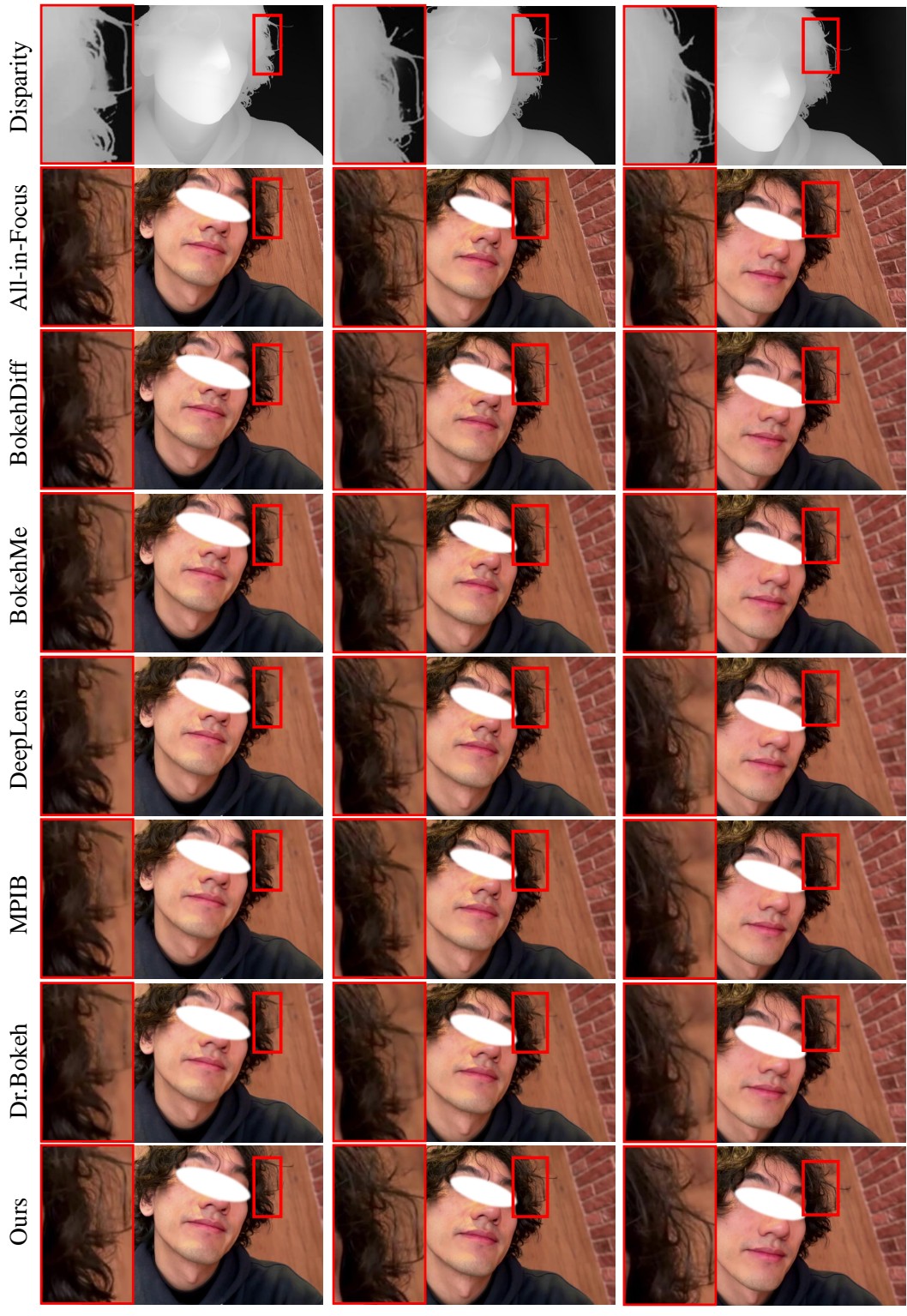

Figure 12: Comparison results with baselines on real-world videos. The area inside the red border is zoomed in to highlight more details. Please zoom in to view them.

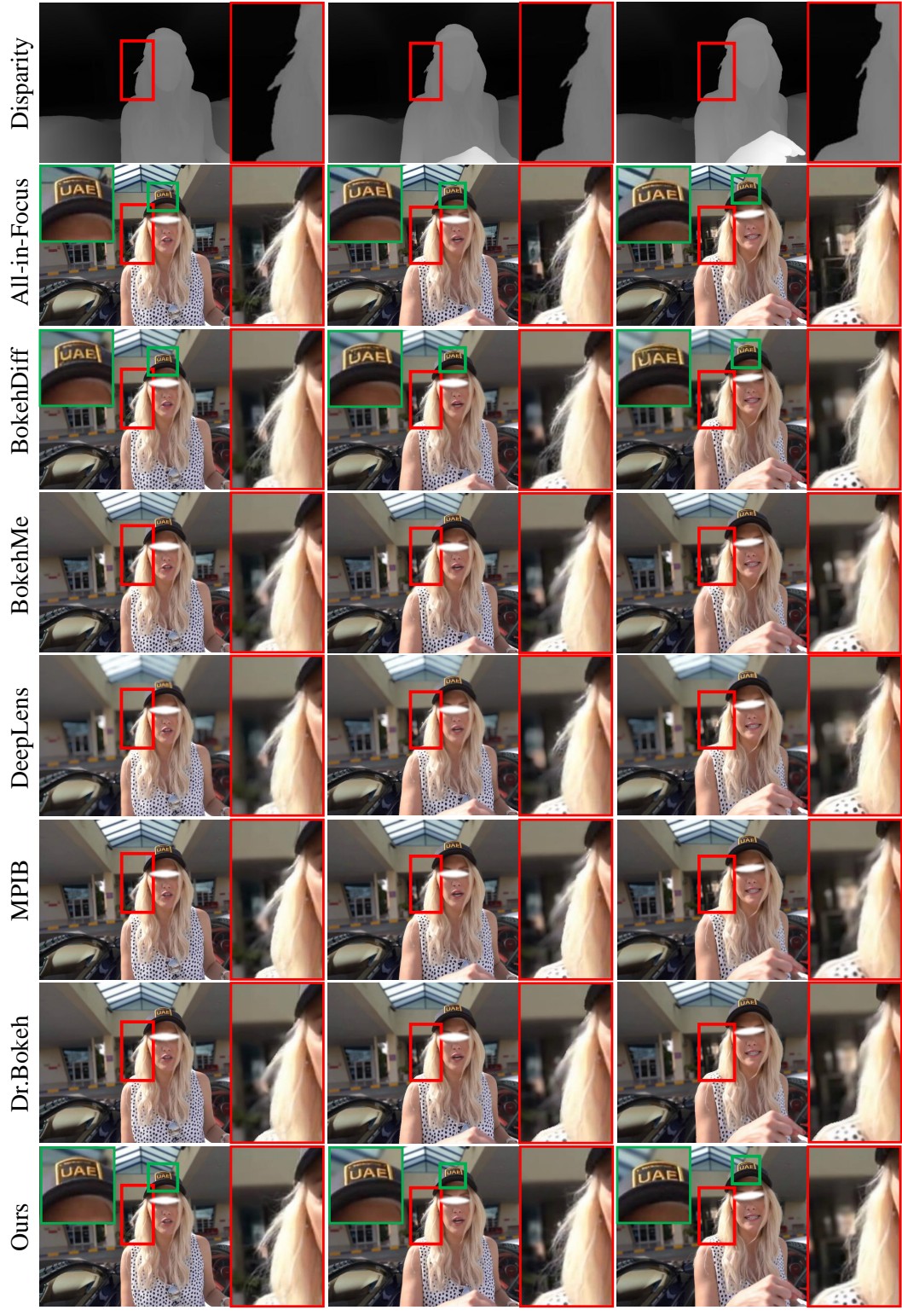

Figure 13: Comparison results with baselines on real-world videos. The area inside the red and green border is zoomed in to highlight more details. Please zoom in to view them.

