# OpenReview forum: "Any-to-Bokeh: Arbitrary-Subject Video Refocusing with Video Diffusion Model"
_ICLR.cc/2026/Conference — ICLR 2026 Poster_

### Official Review · Reviewer_cW3t · 2025-10-26

**Soundness:** 2
**Presentation:** 3
**Contribution:** 2
**Rating:** 4
**Confidence:** 4

**Summary:**

In this paper, the authors present a video bokeh rendering method based on diffusion model. They propose a one-step diffusion framework, and employ a multi-plane image representation to the focal plane as a condition. Also, they introduce a progressive training strategy for stability. Experiments on synthetic and real-world benchmarks show the performance of their methods.

**Strengths:**

- The authors propose a one-step diffusion framework for video bokeh rendering, which exhibits an efficiency advantage in inference time.
- In the third stage, fine tune the VAE decoder and introduce texture loss based on image gradients to improve high-frequency texture and edge clarity, which helps ensure the presentation of details of the focused subject.
- Time consistency and video quality indicators are significantly better than the baseline mentioned in the paper.

**Weaknesses:**

- Comparison with Video Bokeh Methods. The authors only compared their with the image bokeh method, but it needs to be compared with video methods, such as VBR [1].
- Complex scenes. The authors employ a multi-plane image (MPI) representation, and this representation can bring challenges, such as whether to divide an object into two different layers. The authors should discuss this situation.
- The robustness of this method. The authors should compare their full model with degraded depth maps and without degraded depth maps.
- Missing details. TS (token selection) is not defined in formula (4); How M̄=[1,M] aligns a “global token” with 2D masks should be provided; Specify mask resolutions for each block and the exact interpolation strategy; clarify how “near-focus vs. wide-interval” masks are selected per layer.
- Dataset and generalization. Both synthetic training and testing are based on the construction of "planar disparity" (where d is an affine function of x and y). This simplifies geometry but differs significantly from real-world scenarios.

[1] Luo, Yawen, et al. "Video bokeh rendering: Make casual videography cinematic." Proceedings of the 32nd ACM International Conference on Multimedia. 2024.

**Questions:**

- Weighted overlap inference strategy. I wonder if doing weighted overlap only during inference without the same operation during training will affect inference performance.
- The number of image plane N. I want to know whether to use the same N or different N for different scenarios. If different, how is the value of N determined.
- Additional metrics. Add LPIPS for frame-level perceptual quality;
- Dynamic effects. I suggest the author provide video effects

---

> ### Author Response · Authors · 2025-11-21
> **Rebuttal to reviewer cW3t (part1)**
>
> Thank you for the thoughtful and constructive review. We appreciate your time and insights; they helped us clarify the paper and fix the noted issues in the revision. All corresponding revisions have been marked in **purple** in the updated manuscript.
>
> ---
>
> ### W1: Comparison with Video Bokeh Methods, such as VBR
> We agree that VBR is the first work in this field and should be used as a baseline for comparison. Unfortunately, VBR has never been open-sourced. Before submission and again during the rebuttal period, we contacted the authors to request access to their code and pretrained weights, but they informed us that these resources are no longer available due to the main author’s graduation.
>
> To the best of our knowledge, VBR is the only existing video bokeh baseline, and we explicitly acknowledge the lack of a direct comparison as a limitation in **line 403 (page 8)** of the original submission. As a result, our experiments focus on comparisons with state-of-the-art image-based bokeh methods, which represent the strongest publicly available baselines to validate the effectiveness of our model. We also plan to release our code and trained models to facilitate future comparisons and further research on video bokeh rendering.
>
> ---
>
> ### W2: The authors employ a multi-plane image (MPI) representation, and this representation can bring challenges, such as whether to divide an object into two different layers. The authors should discuss this situation.
>
> In our method, an object being distributed across multiple MPI layers does not introduce the typical artifacts associated with rigid layer assignment. This is because our model does not rely on the MPI representation alone. The key conditioning signal is the disparity sequence, which provides a continuous and precise depth cue for our model.
>
> Our MPI serves as an auxiliary representation designed to allocate finer depth resolution near the focal plane and coarser resolution elsewhere. This adaptive discretization preserves detail where it matters most, but it does not enforce hard object-to-layer boundaries. As a result, even if different parts of an object fall into adjacent layers, the model guided by continuous disparity—naturally maintains geometric and visual consistency.
>
> ---
>
> ### W3: Comparison of results for model at degraded and non-degraded depths
>
> We appreciate the reviewer’s suggestion. We evaluate it by adding progressively stronger noise to the disparity maps. As shown in **Fig. 6 (page 9)**, the mode I setting corresponds to the non-degraded depth input, while larger noise mode represent increasingly degraded depth. Our method maintains stable PSNR as the noise level increases and consistently outperforms all baselines. It achieves the lowest FVD across all perturbation modes and maintains the lowest FD with minimal variation, whereas the baselines degrade noticeably and exhibit a clear upward trend in FD. We describe this experiment in detail in lines **lines 424–431 (page 9)** of the revised manuscript, where the new text is highlighted in red for clarity.
>
> In addition, **Figs. 10 (page 19) and 11 (page 20)** provide qualitative comparisons under suboptimal depth inputs. Our method produces visibly sharper and more accurate object boundaries than the baselines.
>
> ---
>
> ### W4-1: Three questions in Eq. 4
> $$
> \hat{\mathbf{Q}}=\mathbf{Q}+\tanh(\gamma)\cdot
> \mathrm{TS}\left(
> \mathrm{Attn}\big([\mathbf{Q}+\Phi_M(E(\mathbf{K})),\Phi_A(\mathbf{V}_A)],\bar{\mathcal{M}}\big)
> \right)
> $$
> 1. *Token-selection operator $\mathrm{TS}$ details:* $\mathrm{TS}(\cdot)$ selects only the output tokens corresponding to the query tokens $\mathbf{Q}$ after the attention operation, discarding the auxiliary tokens $\mathbf{V}_A$.
> 2. *How $\bar{\mathcal{M}}$ aligns a “global token” with 2D masks should be provided:* $[\cdot,\cdot]$ denotes concatenation, and $\bar{\mathcal{M}}=[\mathbf{1},\mathcal{M}]$ denotes the mask augmented by padding an all-ones matrix, which is used in the masked-attention computation.
> 3. *Specify mask resolutions for each block and the exact interpolation strategy:* Our U-Net consists of four downsampling layers and four upsampling layers with feature resolutions of $[72,128],[36,64],[18,32],[9,16]$. We apply bilinear interpolation to generate masks at the appropriate resolutions.
>
> These clarifications have been incorporated into the revised manuscript in **lines 232 and 235 (page 5)** to improve readability.

---

> ### Author Response · Authors · 2025-11-21
> **Rebuttal to reviewer cW3t (part2)**
>
> ### W5: There is a gap between the geometry of our synthetic dataset and real-world scenarios
> We acknowledge that there is an inherent geometry gap between our synthetic dataset and real-world scenes. This is mainly due to the absence of publicly available datasets with paired all-in-focus and bokeh videos; therefore, our synthetic dataset serves as a practical substitute.
>
> To mitigate this gap, we build on a large-scale pre-trained video diffusion model whose geometry priors help the model generalize to real scenes and maintain structural consistency.
>
> In practice, the model trained only on our synthetic data achieve state-of-the-art performance on both synthetic and real-world test sets (**Table 1 in page 7 and Table 3 in page 9**), and they significantly reduce temporal flickering and inconsistent blur transitions compared to image-based methods (**Figs. 4 in page 7 and 10–13 in page 19-22**).
>
> ---
>
> ### Q1: If weighted overlap inference strategy is applied only at inference and not during training, does it degrade performance?
>
> In this work, we apply the Weighted Overlap Inference Strategy (WOIS) exclusively at inference time, not during training.
>
> Applying the weighted overlap inference strategy (WOIS) only at inference does not degrade performance or introduce any train–test mismatch. WOIS does not modify the model parameters or its internal predictions; it is a parameter-free post-processing step that performs deterministic weighted averaging in the overlapping regions between neighboring segments to avoid flickering and misaligned blur transitions at segment boundaries.
>
> Empirically, this is supported by our ablation in **Table 5 (page 10)**. Comparing the second and third rows, applying WOIS only at inference consistently improves image quality (SSIM, PSNR, LPIPS), video quality (FVD, VFID-I), and temporal consistency (FD). The corresponding numbers are summarized in `Table R1`, confirming that WOIS is beneficial rather than harmful.
>
> **Table R1: Ablation study of WOIS.**
>
> | WOIS | FD (↓) | RM (↓) | VFID-I (↓) | FVD (↓) | SSIM (↑) | PSNR (↑) | LPIPS (↓) |
> |:----:|:------:|:------:|:-----------:|:-------:|:--------:|:--------:|:--------:|
> | ✓    | 0.540  | 0.013  | 4.209       | 20.743  | 0.905    | 32.035   | 0.052 |
> | -    | 0.551  | 0.013  | 4.521       | 21.941  | 0.905    | 31.936   | 0.054 |
>
> ---
>
> ### Q2: How is the number of image plane N determined
>
> In our implementation, we set the number of image planes to $N=4$ to align with the four downsampling blocks of the U-Net of Stable Video Diffusion. This allows each MPI plane to be associated with a different feature scale in the hierarchy. We have added this clarification around **line 312 (page 6)** in the revised manuscript.
>
> ---
>
> ### Q3: Add LPIPS for frame-level perceptual quality
> We thank the reviewer for the suggestion. We have added frame-level LPIPS as an additional perceptual quality metric in **Tables 1, 5, 6, and 7 (pages 7 and 10)**, and reproduce the updated parts in `Tables R2–R5` for clarity.
>
> After including this metric, all conclusions in the paper remain unchanged: our method still achieves state-of-the-art performance compared with other baselines. Moreover, across all ablation studies, our proposed settings continue to deliver the best results.
>
> **Table R2: Quantitative comparison of Any-to-Bokeh. The best metric scores in each column are marked in bold for clarity.**
> | Metric | DeepLens | BokehDiff | BokehMe | Dr.Bokeh | MPIB  | Ours  |
> |--------|----------|-----------|---------|----------|-------|-------|
> | LPIPS↓ | 0.183    | 0.127     | 0.060   | 0.046    | 0.040 | **0.019** |
>
> **Table R3: Ablation study of Any-to-Bokeh module. ''MPI'': MPI spatial block. ''OS'': one-step inference schedule. ''WOIS'': weighted overlap inference strategy. ''TR'':  temporal refinement.**
> | Variants | MPI | OS | WOIS | TR | LPIPS↓ |
> |----------|-----|----|------|----|--------|
> | #1       | ✓   | ✓  | ✓    | ✓  | 0.051  |
> | #2       | ✓   | ✓  | ✓    | -  | 0.052  |
> | #3       | ✓   | ✓  | -    | -  | 0.054  |
> | #4       | ✓   | -  | -    | -  | 0.059  |
> | #5       | ✓   | -  | -    | -  | 0.069  |
>
> **Table R4: Ablation study on FV**
> | FV | LPIPS↓ |
> |----|--------|
> | ✓  | 0.019  |
> | -  | 0.051  |
>
> **Table R5: Ablation study on TR under noisy depth.**
> | TR | LPIPS↓ |
> |----|--------|
> | ✓  | 0.050  |
> | -  | 0.054 |
>
> ---
>
> ### Q4: I suggest the author provide video effects
> We have included multiple video examples in the **supplementary material**, which can be downloaded for viewing. For convenience, the videos are organized in the folder corresponding to your reviewer ID (**reviewer cW3t**). These examples showcase three key application scenarios: customizable focal plane, adjustable blur strength, and long-term video bokeh rendering, and also provide visual comparisons with baseline methods, demonstrating our advantages in boundary fidelity and temporal consistency.

---

### Official Review · Reviewer_7dEg · 2025-10-28

**Soundness:** 4
**Presentation:** 4
**Contribution:** 3
**Rating:** 8
**Confidence:** 3

**Summary:**

The authors propose a method for adding controllable Bokeh in videos. They fine tune a video diffusion model to accept a video and a corresponding explicit scene geometry conditioning and output a depth-aware Bokeh video in a single diffusion step. The authors propose a multi-stage training strategy that facilitates robustness to noisy input geometry and high temporal bokeh consistency. The authors provide extensive evaluations showing state-of-the-art results for controllable video bokeh, with control of the focus plane and bokeh intensity.

**Strengths:**

1.	The authors leverage the prior of a video diffusion model in a novel way, to perform temporally consistent, controllable video bokeh.
2.	The authors showcase good bokeh results. They also provide a supplementary video with their results and comparisons to other methods, which is very important for the qualitative assessment of their claims for temporally consistent bokeh addition.
3.	The authors show extensive quantitative evaluations, emphasizing their lead over other competing methods.
4.	The authors provide several ablations for their training strategy choices.

**Weaknesses:**

Major:
1.	The authors do not provide limitations for their method. Are there any scenarios where the model fails to generate a good bokeh video? Maybe in videos with fast motion, such as a car race.
Minor:
1.	Figure 1 is not referenced.
2.	SM figures 10,11 – red border not corresponding to zoom-in area.

**Questions:**

Please see weaknesses section.

---

> ### Author Response · Authors · 2025-11-21
>
> Thank you for your positive evaluation of our contribution to the paper. We also sincerely appreciate your constructive comments and suggestions. All corresponding revisions have been marked in **yellow** in the updated manuscript.
>
> ---
>
> ### W1: The authors do not provide limitations for their method
> While Any-to-Bokeh achieves significant improvements over existing methods, it still has two main limitations. First, due to limited computational resources, we did not investigate training with larger batch sizes. With more GPUs and a larger-scale training setup, the model could further improve its ability to handle complex scenes. Second, for long videos, we cannot process the entire sequence in a single forward pass; instead, inference relies on repeatedly applying the WOIS strategy over overlapping segments, which sacrifices some efficiency. We add this discussion in **Appendix A.7 (page 19)** and treat these limitations as key directions for future work.
>
> ---
>
> ### W2: Can the model work in fast-moving scenarios such as racing?
> Our model still produces satisfactory results in this scenario. We present the visualizations in **Fig. 9 (page 18)** of the appendix and provide the corresponding video files in the **supplementary material**. You can view them by opening the folder associated with your reviewer ID (**reviewer 7dEg**).
>
> ---
>
> ### W3: Figure 1 is not referenced. Figures 10,11 – red border not corresponding to zoom-in area.
>
> In the revised manuscript, we have added an explicit reference to Fig. 1 at **lines 89–91 (page 2)**. We have also corrected the misalignment between the red borders and the zoomed-in regions in Figs. 10 and 11; the updated versions are now provided as **Figs. 12 and 13 (pages 21–22)**.

---

### Official Review · Reviewer_Y63a · 2025-10-30

**Soundness:** 3
**Presentation:** 3
**Contribution:** 3
**Rating:** 6
**Confidence:** 3

**Summary:**

This paper introduces a novel one-step diffusion framework for generating temporally coherent and depth-aware video bokeh effects. The proposed approach enables video refocusing with explicit control over the focal plane and bokeh intensity, addressing the limitations of prior image-based and video-based bokeh rendering methods. The framework leverages a focal-plane-adapted multi-plane image (MPI) representation to guide the diffusion process, ensuring temporal smoothness and accurate depth-dependent blur transitions.

**Strengths:**

1. **Innovative Approach:**
   The use of an MPI-guided conditioning mechanism in a one-step diffusion framework for video bokeh generation is both original and well-motivated. It effectively bridges the gap between static image refocusing and temporally coherent video refocusing.

2. **Temporal Coherence and Depth Awareness:**
   The focal-plane-adapted MPI representation efficiently balances detail preservation in focused regions and smooth transitions in defocused areas, improving visual consistency across frames.

3. **Comprehensive Experiments:**
   The paper includes thorough quantitative and qualitative comparisons with existing methods, demonstrating consistent improvements in temporal stability, spatial accuracy, and controllability.

**Weaknesses:**

1. **Dependence on Depth Estimation:**
   The method relies on pre-trained depth estimation models as input. In dynamic or complex scenes, depth errors may propagate into the final bokeh rendering. The paper would be stronger with a sensitivity analysis or ablation showing how depth inaccuracies affect output quality.

2. **Computational Efficiency:**
   While the results are impressive, the paper provides limited discussion on computational cost. Diffusion-based models are typically resource-intensive; more details on runtime, memory consumption, and scalability (e.g., potential for real-time use) would enhance the practical relevance.

3. **Dataset Bias and Generalization:**
   The primary evaluation uses synthetic datasets. Although this allows for controlled comparisons, such datasets may not fully reflect real-world complexities such as fast motion, varying lighting, or occlusions. Additional experiments on diverse real-world datasets would strengthen claims of robustness and generalizability.

**Questions:**

See weaknesses above.

---

> ### Author Response · Authors · 2025-11-21
>
> We sincerely thank you for your positive and encouraging feedback, as well as the valuable insights provided in your evaluation and revision suggestions. All corresponding changes have been marked in **red** in the updated manuscript.
>
> ---
>
> ### W1: The paper would be stronger with a sensitivity analysis or ablation showing how depth inaccuracies affect output quality.
>
> In the revised manuscript, we add a dedicated sensitivity analysis that evaluates our model under depth inaccuracies by injecting progressively stronger noise into the disparity maps. As shown in **Fig. 6 (page 9)**, increasing noise leads to performance degradation for all methods. However, our method maintains stable PSNR as the noise level increases and consistently outperforms all baselines. It achieves the lowest FVD across all perturbation modes and maintains the lowest FD with minimal variation, whereas the baselines degrade noticeably and exhibit a clear upward trend in FD. We describe this experiment in detail in **lines 424–431 (page 9)**.
>
> In addition, **Figs. 10 (page 19) and 11 (page 20)** provide qualitative comparisons under suboptimal depth inputs. Our method produces visibly sharper and more accurate object boundaries than the baselines.
>
> ---
>
> ### W2: The paper provides limited discussion on computational cost.
>
> Thank you for your positive assessment of our model’s performance. As a diffusion-based approach, our method indeed incurs non-trivial computational cost. In the revised manuscript, we provide a detailed comparison of computational cost with all baselines in **Table 4 (page 9)** and reproduce the corresponding numbers in `Table R1`.
>
> All methods are evaluated on single-frame inputs at a resolution of 576×1024 on the same H800 GPU to ensure fairness. For BokehMe and Dr.Bokeh, which rely on custom CUDA-based renderers, full FLOPs and parameter counts are not directly comparable. Therefore, for BokehMe, we report only the neural network component (marked with “∗”). Despite using a larger backbone, our model achieves the shortest runtime among all methods. This efficiency stems from our end-to-end optimization and a highly parallelizable architecture. Compared with the diffusion-based method BokehDiff, our approach is more efficient in terms of parameter count and VRAM usage, and it also offers a clear advantage in inference time over the traditional MPI-based method MPIB and Dr.Bokeh. We have added this discussion to the computational cost section in the revised paper in **line 453-465 (page 9)**.
>
>
> **Table R1: Results of the computational cost comparison on single frame.**
> | Method | Time (s) | Params (M) | GFLOPs | VRAM(GB) |
> |--|--|--|--|--|
> | BokehMe  | 0.103| ∗1.41| ∗169.71| 1.4|
> | MPIB     | 0.521| 27.46| 1418.50| 3.3|
> | Dr.Bokeh | 2.729| —|—| 2.7|
> | BokehDiff| 0.799| 2459| 3430| 18.4|
> | Ours     | 0.094| 1880| 3620| 13.6|
>
> ---
>
> ### W3: Additional experiments on diverse real-world datasets such as fast motion, varying lighting, or occlusions
>
> Indeed, fast motion, varying lighting, and occlusions pose significant challenges for bokeh rendering. We have tested our method on scenarios involving fast motion (fast_motion.mp4), varying lighting (varying_lighting.mp4), and occlusions (occlusion.mp4). Our method still produces high-quality bokeh rendering. The corresponding videos have been updated in the **supplementary material**, which you can access by clicking the folder associated with your ID (**reviewer Y63a**). We also include these visualization results in the revised **Fig. 9 (page 18)**.
>
> Furthermore, we validated our method on the real-world DAVIS dataset. To evaluate performance on this dataset, we introduced the VEPI metric (described in Appendix A.4 in page 17), which measures the model’s ability to preserve detail at the edges of the focused subject. As reported in **Table 1 (page 7)**, our method achieves state-of-the-art performance. For your convenience, we reproduce these results in `Table R2`.
>
> **Table R2: Results of the computational cost comparison.**
>
> | Metric | DeepLens | BokehDiff | BokehMe | Dr.Bokeh | MPIB | Any-to-Bokeh |
> |------|------|------|------|------|------|----------|
> | VEPI↑| 0.715| 0.859| 0.937| 0.863| 0.921| **0.944**|
>
> In addition, we conducted a user study on the DAVIS dataset. We randomly selected 20 videos and rendered them using different methods. In each trial, participants were shown a pair of videos: one generated by our method and one by a randomly chosen baseline. Participants were asked to select the one with the more aesthetically pleasing bokeh effect based on their personal preference. The results are reported in **Table 3 (page 9)** of the paper and are reproduced in `Table R3` below for your convenience.
>
> **Table R3: Results on human preference.**
>
> | Baseline| Preference|
> |---|---|
> | Ours vs. DeepLens | 96.9% / 3.1% |
> | Ours vs. BokehMe  | 77.1% / 22.9%|
> | Ours vs. MPIB     | 62.9% / 37.1%|
> | Ours vs. Dr.Bokeh | 77.8% / 22.2%|
> | Ours vs. BokehDiff| 75.7% / 24.3%|

---

### Author Response · Authors · 2025-12-01

Dear Reviewers, ACs, SACs, and PCs,

We sincerely thank all reviewers for their time and thoughtful feedback on our manuscript. We are grateful for the constructive comments and the positive recognition of our contributions.

We are encouraged that Reviewers **Y63a** and **7dEg** share our focus on temporally consistent and controllable video bokeh, and that Reviewer **cW3t** recognizes the high video quality and pleasing bokeh produced by our method.

**All reviewers** expressed interest in how our model behaves in more challenging real-world scenarios, such as fast motion, varying lighting, and occlusions. In response, we have provided additional qualitative results for these cases in Fig. 9 (page 18) and included the corresponding videos in the supplementary material, where our model consistently produces high-quality bokeh rendering even under these complex conditions.

Reviewers **Y63a** and **cW3t** also asked for a deeper analysis of the model’s robustness to depth inaccuracies and its generalization to real-world data. To address this, we added a sensitivity analysis in Fig. 6 (page 9), showing performance under different levels of depth perturbation, and qualitative comparisons under suboptimal depth inputs in Figs. 10 (page 19) and 11 (page 20). Furthermore, we evaluated our method on the real-world DAVIS dataset, where it achieves state-of-the-art performance as reported in Table 1 (page 7). In addition, a user study summarized in Table 3 (page 9) further supports the perceptual advantages of our approach.


Following Reviewer **cW3t**’s suggestions, we have also included LPIPS in the main tables as a frame-level perceptual quality metric, and our method remains the top-performing approach under this measure. In addition, we engaged in detailed discussions with Reviewer cW3t regarding the model design. We clarified that (1) our proposed MPI representation is designed to allocate finer depth resolution near the focal plane and coarser resolution elsewhere, serving as an auxiliary mechanism for injecting depth information and thereby improving the model’s performance; (2) we have further specified the implementation details of Eq. 4 in the revised manuscript; and (3) our weighted overlap inference strategy (WOIS) does not introduce any additional trainable parameters and therefore does not cause a train–test mismatch, and our ablation study shows that applying WOIS at inference time consistently improves performance.

The professionalism and insightful feedback from all reviewers have significantly improved the clarity and rigor of our work. We once again express our sincere gratitude to all reviewers, as well as to the ACs, SACs, and PCs, for their time and dedicated effort throughout this review process.

---

### Meta-Review · Area_Chair_oUqL · 2026-01-06

**Summary:**

The paper proposes a framework for generating controllable video bokeh effects using a video diffusion model. The core technical novelty lies in conditioning the diffusion model with a focal-plane-adapted Multi-Plane Image (MPI) representation, which allows for better utilization of 3D priors from pre-trained backbones (specifically SVD). The method also introduces a progressive training strategy and a Weighted Overlap Inference Strategy (WOIS) to ensure temporal coherence and handle arbitrary video lengths.

Overall, the reviewers found the approach innovative, particularly the bridge it builds between geometric consistency (via MPI) and generative quality (via Diffusion). The results demonstrate state-of-the-art performance in temporal stability and boundary handling compared to existing image-based or traditional rendering methods. While there were initial concerns regarding robustness to depth estimation errors and reliance on synthetic data, the rebuttal provided substantial evidence to alleviate these worries. I agree with the consensus that this work establishes a solid baseline for video bokeh generation.

**Reviewer Concerns:**

**Addressed Concerns**

(1) Robustness to Depth Inaccuracies (Reviewers Y63a, cW3t). Both reviewers were concerned that the method's reliance on depth estimation would lead to artifacts when depth maps are imperfect. The authors included a new sensitivity analysis (Fig. 6 in the revised paper) and qualitative results (Figs. 10-13) showing the model maintains stability even with significant noise added to disparity maps. I find this evidence convincing.

(2) Generalization to Real-World Scenarios (Reviewers Y63a, 7dEg, cW3t). There was skepticism about training on synthetic data and testing on real videos. The authors addressed this by providing results on the DAVIS dataset, including specific "stress tests" requested by Reviewer 7dEg (fast motion, occlusions, varying lighting). The user study results further support the perceptual quality on real-world data.

(3) Computational Cost (Reviewer Y63a). The authors provided a detailed breakdown (Table 4), showing that their method is actually faster than some traditional baselines (Dr. Bokeh) and significantly more efficient than competing diffusion-based methods (BokehDiff), addressing the efficiency concern.

(4) Technical Clarifications (Reviewer cW3t). The reviewer raised questions about the specific implementation of the token selection (Eq. 4) and the handling of object layers in MPI. The authors clarified the mathematical details and the auxiliary nature of the MPI representation in the text.

**Outstanding / Minor Concerns**

Comparison with Video Baselines (Reviewer cW3t). Reviewer cW3t noted a lack of comparison with "VBR" (Luo et al., 2024). The authors explained that VBR is not open-source and the code is unavailable. As an AC, I accept this justification; authors cannot be penalized for failing to compare against inaccessible methods. The comparison against strong image-based baselines adapted for video is sufficient for this stage of research.

**Reviewer Scores:**

Reviewer 7dEg (Current: 8->8). This reviewer was already very positive ("Excellent"). The authors addressed their minor concerns regarding limitations and formatting.

Reviewer Y63a (Current: 6->6). This reviewer's main hesitation was the lack of sensitivity analysis regarding depth errors and computational cost details. Since the authors provided a comprehensive sensitivity ablation and cost table, the "marginal" nature of the acceptance is removed.

Reviewer cW3t (Current: 4->4/6). This reviewer was the most critical, focusing on the lack of VBR comparison and technical specifics. The authors provided valid reasons for the missing VBR comparison and clarified the technical components (WOIS, Eq. 4). The reviewer explicitly noted they "would not mind if the paper is accepted." Given the clarifications and additional LPIPS metrics provided.

---

### Decision · Program_Chairs · 2026-01-26

Accept (Poster)